# Fluorescent Silica Nanoparticles Targeting Mitochondria: Trafficking in Myeloid Cells and Application as Doxorubicin Delivery System in Breast Cancer Cells

**DOI:** 10.3390/ijms23063069

**Published:** 2022-03-12

**Authors:** Federica Sola, Mariele Montanari, Mara Fiorani, Chiara Barattini, Caterina Ciacci, Sabrina Burattini, Daniele Lopez, Alfredo Ventola, Loris Zamai, Claudio Ortolani, Stefano Papa, Barbara Canonico

**Affiliations:** 1Department of Biomolecular Sciences, University of Urbino Carlo Bo, 61029 Urbino, Italy; f.sola@campus.uniurb.it (F.S.); mariele.montanari@uniurb.it (M.M.); mara.fiorani@uniurb.it (M.F.); c.barattini@campus.uniurb.it (C.B.); caterina.ciacci@uniurb.it (C.C.); sabrina.burattini@uniurb.it (S.B.); d.lopez1@campus.uniurb.it (D.L.); loris.zamai@uniurb.it (L.Z.); claudio.ortolani@uniurb.it (C.O.); stefano.papa@uniurb.it (S.P.); 2AcZon s.r.l., 40050 Monte San Pietro, Italy; alfredoventola@aczonpharma.com; 3Department of Pure and Applied Sciences (DiSPeA), University of Urbino Carlo Bo, 61029 Urbino, Italy

**Keywords:** nanoparticles, doxorubicin, mitochondria, lysosomes, anticancer tool, theranostics, delivery system, EVs

## Abstract

Fluorescent silica nanoparticles (SiNPs) appear to be a promising imaging platform, showing a specific subcellular localization. In the present study, we first investigated their preferential mitochondrial targeting in myeloid cells, by flow cytometry, confocal microscopy and TEM on both cells and isolated mitochondria, to acquire knowledge in imaging combined with therapeutic applications. Then, we conjugated SiNPs to one of the most used anticancer drugs, doxorubicin (DOX). As an anticancer agent, DOX has high efficacy but also an elevated systemic toxicity, causing multiple side effects. Nanostructures are usually employed to increase the drug circulation time and accumulation in target tissues, reducing undesired cytotoxicity. We tested these functionalized SiNPs (DOX-NPs) on breast cancer cell line MCF-7. We evaluated DOX-NP cytotoxicity, the effect on the cell cycle and on the expression of CD44 antigen, a molecule involved in adhesion and in tumor invasion, comparing DOX-NP to free DOX and stand-alone SiNPs. We found a specific ability to release a minor amount of CD44+ extracellular vesicles (EVs), from both CD81 negative and CD81 positive pools. Modulating the levels of CD44 at the cell surface in cancer cells is thus of great importance for disrupting the signaling pathways that favor tumor progression.

## 1. Introduction

Efforts in biomedical and pharmaceutical nanotechnology have driven a high number of studies on the biological effects of nanomaterials. In our previous work [1], we reported a physicochemical characterization of fluorescent silica nanoparticles (SiNPs), interacting with biological models (U937 and PBMC cells), describing the specific triggered biologic response. We observed that fluorescent silica nanoparticle entry involves lysosome and mitochondria pathways, highlighting a SiNP colocalization. Despite the main mitochondria localization, fluorescent silica NPs did not induce a significant increase of intracellular ROS, known inductors of apoptosis, during the time course of analyses. We analyzed in depth the preferential mitochondria targeting, with the goal of acquiring knowledge in combining imaging and localized therapeutic applications in a unique tool. Interestingly, we observed an almost complete colocalization of the nanoparticles with mitochondria, about 2-fold higher than lysosomal colocalization. Due to the vital role of apoptosis, mitochondria were considered to be a potential drug delivery target. Mitochondria dysfunction attributed to targeted drug delivery of a therapeutical agent to mitochondria could not only cutoff the “energy supply”, which was necessary to P-glycoprotein (P-gp) function [2], but also induce the apoptosis cascade process and sufficiently overcome multidrug resistance (MDR) [3,4]. A suitable NP system may improve the efficacy of chemotherapy, using the pathophysiology of tumors, especially the enhanced permeability and retention (EPR) effect and delivering drugs to the target through an active targeting strategy due to the receptor-mediated endocytosis mechanisms [5,6,7]. Owing to their small size, NPs have a great ability to pass through biological barriers, such as the pulmonary system and the tight junctions of endothelial cells of the skin, thus increasing the systemic bioavailability of the drugs [8].

In the present study, we examined in-depth the localization of fluorescent silica nanoparticles in U937 cells and their effects on isolated mitochondria, providing insights about the effects of DOX-NPs on the breast cancer cell line MCF-7, compared to free DOX and stand-alone SiNPs, since DOX is currently one of the most effective agents in the treatment of breast cancer [9]. Generally, NP-based drug delivery systems are constructed by the incorporation of active drugs via entrapment, conjugation or encapsulation into NPs, depending on the peculiar characteristics of the system [10]. To this extent, firstly, we changed the amine functional groups (-NH2) present on the NP shell to carboxyl groups (-COOH) in order to bind the drug.

Doxorubicin (DOX) is an antibiotic deriving from the anthracycline family that is widely employed in breast cancer therapy. The proposed mechanism of action includes the intercalation into DNA, disrupting gene expression and macromolecule biosynthesis, production of reactive oxygen species (ROS) and inhibition of topoisomerase II (topo II), a key enzyme in DNA synthesis and replication [5,11]. Nevertheless, one of the most essential problems of DOX-based chemotherapy is the lack of specificity to target cells and the involvement of healthy tissues as well. Like all the other anticancer agents, DOX has a high efficacy but also an elevated systemic toxicity and causes multiple side effects [5,12]. Furthermore, the development of drug resistance in DOX treatment leads to the failure of this therapy. The major molecular mechanism responsible for the anticancer activity of DOX includes DNA lesions [13], induction of apoptosis by activation of p53 protein and failure of mitochondrial function. It has been shown that the cellular DOX accumulation increases production of reactive oxygen species (ROS) and triggers damage to mitochondrial respiration related to protein synthesis and lipid peroxidation, release of cytochrome c from mitochondria, and activation of caspases [14,15]. Several research groups have worked on the development of NPs for DOX delivery and many systems are already approved by the FDA, most of which are based on liposomes (e.g., Doxil^®^, Lipodox^®^ or Myocet^®^) [16]. Fluorescent silica nanoparticles (SiNPs) synthetized through a micelle-assisted method, as presented by Pellegrino et al. [17], exhibited exciting results in term of cytotoxicity and intracellular localization. SiNPs showed interesting features, mostly regarding the versatility of the technology, which makes them potentially interesting for several applications, including drug delivery, in the pool of theranostic techniques. SiNPs are core-shell systems where the core is a silica matrix in which are entrapped fluorophores and one of the most used “stealth” polymers in the drug delivery field, polyethylene glycol (PEG), comprises the shell, which is the first interaction with the external environment and allows the modulation of functional groups exposed on the NP surface. Here, on MCF-7 cells, we assessed the first step by conjugating a therapeutic molecule to a well-known imaging platform. In future, other steps will be necessary to add a targeting molecule, by exploiting the functional groups on the shell surface, in order to achieve a specific cellular targeting and intracellular drug delivery via receptor-mediated endocytosis.

## 2. Results and Discussion

### 2.1. In-Depth Analysis of Fluorescent SiNPs Localization and Effects on Isolated Mitochondria

To deeply characterize mitochondria uptake of SiNPs, and their status after their internalization, we isolated mitochondria from control and SiNPs-treated U937 cells, by means of the protocol highlighted in the Section 4. Isolated mitochondria were tested by flow cytometry and confocal microscopy.

In Figure 1A, brightfield images of isolated mitochondria from controls (1 h and 24 h) and NP-treated cells (1 h and 24 h) are visible. In these images we can appreciate a slight reduction in the size of this organelle. Furthermore, fluorescence colocalization highlights rounded, yellow/green, MTG/TMRE labeled mitochondria, with onion-shaped cristae, from control, whereas NP treated cells evidence a light violet fluorescence (originating by the overlapping of red and green colors with the blue of SiNPs) (Figure 1B). The fluorescence images show the peripheral distribution of SiNPs, located in the peri-mitochondrial space, underlining that the inner membrane is a valid barrier against delivery of exogenous material into the mitochondrial matrix. The SiNPs positive mitochondria are more elongated and seem polarized, suggesting the budding of mitochondrial derived vesicles: however, these aspects will be clarified by TEM analyses.

The quantitative flow cytometric analyses show a 2-fold increase of mitochondria that are SiNPs positive, (40% after 1 h vs 80% after 24 h) (Figure 1C). Furthermore, the SiNPs fluorescence intensities are 3-fold higher after 24 h than after 1 h (Figure 1D), suggesting that in the same cells, SiNPs tend to accumulate in a specific mitochondrion.

The contour plots and histograms in Figure 2 illustrate the sequence of analyses and gating strategy by which we quantitate mitochondrial SiNPs uptake. In controls there is an absence of any fluorescent signal in the NP fluorescent channel, whereas in SiNPs positive samples, the right shift of fluorescence is clear and intense. To establish the healthy condition of isolated mitochondria we also used carbonyl cyanide m-chlorophenylhydrazone (CCCP), which is a typical mitochondrial uncoupler, leading to the dissipation of mitochondrial membrane potential [18]. After the incubation we observed a decrease of TMRE resulting in a correct procedure of mitochondria isolation. However, by inserting MTG and TMRE in this multiparametric panel, we are able to evaluate the mass and function of isolated mitochondria.

Intriguingly, in the contour plot FSC vs. MTG (FITC), in particular after 24 h, a mitochondria population with high MFI MTG values is observable. A subset of this population also displays higher FSC values: this is a further suggestion for mitochondrial fission and/or mitochondrial-derived vesicles (MDVs) budding from the outer membrane of mitochondria.

The increase of both MTG and TMRE (Figure 2 and Figure 3A,B) after 24 h from SiNPs explains the reduction of mitochondrial GSH content (Figure 3D). In fact, the increase in MTG may account for MDV formation and starting the fission process, and it is accompanied by TMRE increase, denoting augmented △Ψm. These sequelae of events lead to inhibition of the levels of mitochondrial glutathione to impair free radical scavenging, leading to further increases in ROS [18]. Importantly, CCCP administration shows that isolated mitochondria are still sensitive to the action of this uncoupler and the behavior of both TMRE and MTG probes is what we expected (Figure 2).

These findings apparently are not in agreement with our previous data on mitochondria, inside cells. However, the isolation procedure could exacerbate some mitochondrial damage or lesion processes. More importantly, however, the total intracellular GSH content increased after both 1 h and 24 h (Figure 3C), leading to ROS buffering and cell homeostasis, since we did not register significant cell death phenomena, until 72 h of investigations.

Our data and the recent references on the field depict the following scenario:(1)Cells after 1 h maintain the ability to synthesize GSH in the cytosol and its transport to mitochondria is not compromised; indeed the transport is increased in respect to control cells, as demonstrated by data on mitochondrial GSH.(2)After 24 h, total GSH of SiNPs treated-cells is higher than control cells and mitochondrial GSH is slightly reduced, suggesting that the NP filled peri-mitochondrial space, may impair specific inner mitochondrial membrane (IMM) carriers, able to transport GSH.(3)MDVs budding (to deeply characterize) should have the capability to partially free mitochondria from SiNPs MDVs and this option is absolutely in agreement with the viable status of the cells and SiNPs release in the environment [1].

### 2.2. TEM Analysis of Different Intracellular Localization: Focus on Mitochondria Behavior

The uptake of SiNPs in mitochondria was demonstrated also by TEM analysis (Figure 4A stained mitochondria, Figure 4B not stained mitochondria), considering the entire cell, without the procedure of mitochondria isolation, to ensure the visibility of mitochondria typical response, such as mitochondria-derived vesicle (MDV) formation. MDVs carry material specifically between mitochondria and other organelles. Mitochondria release small MDVs from their membranes to deliver specific mitochondrial contents to the late endosome/multivesicular bodies for subsequent lysosomal degradation. MDVs have a diameter of 70–150 nm, as it is observable in Figure 4C–E.

Indeed, such analysis confirms a moderate and transient mitochondrial damage (arrows): the known cristae rupture and disappearance [19] is almost absent and their scarce evidences are strictly limited to a restricted area of the mitochondrion (arrows): this event does not represent a key end-point of SiNPs nanotoxicity but the starting point for MDV formation. Presently, MDV trafficking is now included among mitochondrial quality control (MQC) pathways, possibly operating via mitochondrial–lysosomal contacts [20]. In particular, mitochondrial interactions with the endosomal compartment support the shuttle of ions and metabolites across organelles. In Figure 4F,G, the TEM image of this organelle reveals it to be a granular dense core lysosome. Transient or complete fusion between late endosomes and lysosomes results in the formation of endolysosomes. A fine particulate is observable in this organelle, as well as in some mitochondria.

These aspects together with those previously published [1] have been summarized in a schematic illustration (Figure 5) showing all collected data of SiNPs uptake in U937 cells and their intracellular trafficking, by highlighting, particularly, the impact on mitochondria.

### 2.3. Moving to the Doxorubicin Delivery-System: Synthesis and Characterization of DOX-NPs

To complete our study, we decided to obtain fluorescent SiNPs conjugated with doxorubicin (DOX). Fluorescent silica nanoparticles (SiNPs) were synthesized as previously reported by Pellegrino et al. [17], through a micelle-assisted method, where a surfactant, which is an amphiphilic long-chain organic compound able to create micelles in a solution, is used to generate nanoreactors. Besides the silica core, the SiNP shell comprises two different polyethylene glycols (PEGs), both terminating with a trialkoxysilane group: a main component which induces stability and solubility in water, and the counterpart functionalized to have amine reactive groups on the shell. We chose fluorescent SiNPs which are excited by red laser and emit near 660 nm, in order to have absorption spectra of the two species, DOX (λ max Abs 500 nm) and SiNPs, quite far apart and with little influence on each other and, hence, to simultaneously monitor both of them by spectrophotometer and flow cytometer. As depicted in Figure 1, to conjugate DOX to SiNPs, first of all, we modified the functional groups on the shell surface from amine (-NH2) to carboxylic (-COOH) groups with the addition of succinic anhydride. Finally, we enabled COOH-NPs and DOX to react. The drug linked to SiNPs was quantified using a calibration curve drawn linking absorption at 500 nm to the concentration of a set of reference solutions of free DOX in ddH2O and it was calculated to be 25% of therapeutic drug bound to SiNPs.

### 2.4. DOX and NP Uptake and Intracellular Localization

We started from our previous data published in [1] to study the new construct of SiNPs conjugated with DOX. Firstly, we assessed DOX concentrations to be used with MCF-7 cells to have cytotoxic effects. Based on what was reported in literature, we titrated DOX by exposing cells for 24 h to different concentrations of drug and then assessed cell viability through Trypan Blue exclusion assay (Appendix A). We diluted DOX-NPs at the same concentration of drug (1 µg/mL), and, based on the fluorophore concentration of DOX-NPs, we calculated the dilution for COOH-NPs to be used as control and titrated both NP constructs, with the aid of flow cytometry, confirming them to have the same fluorescence intensity (Appendix A). Through confocal analysis, we examined NP localization inside the cells, observing in particular the same organelles considered in the previous work, lysosomes and mitochondria (Figure 6A). After 4 h of incubation, we stained cells with LysoTrackerTM green (LTG), a permeable dye which stains acidic compartments in live cells for lysosomes, whereas for mitochondria TMRE (tetramethylrhodamine ethyl ester perchlorate), which is a cell-permeant and cationic dye sequestered by active mitochondria, was employed. We performed a colocalization analysis, by using JACoP plugin in ImageJ [21]; Pearson’s correlation coefficient (PCC) was used to measure colocalization between SiNPs fluorescence and fluorescence of LTG and TMRE. Concerning COOH-NPs, we detected a higher colocalization with TMRE, as previously observed [1] and detailed above. Of note, DOX-NPs do not show a difference between co-localization with lysosomes or mitochondria (Figure 6C), indicating a possible different intracellular reworking of the two types of nanoparticles. Lysosomes, notably, contain the bulk of hydrolases that, once released into the cytosol, are able not only to induce the apoptotic pathway, but also to amplify the death pathways triggered by other cellular compartments. Furthermore, these organelles receive cargo destined for degradation via the autophagic pathway, by degrading damaged organelles and molecules [1,22]. Thus, the DOX-NP higher colocalization with these organelles may suggest a collaboration with the triggering of cell death or, at longer times, might support the elimination of damaged organelles containing SiNPs, perhaps including mitochondria. Magnifications are depicted in Figure 6B.

Moreover, drug release and accumulation in vitro were monitored through fluorescent microscopy. Thanks to doxorubicin’s characteristic properties, it is notably possible to measure this anticancer drug in biological systems through the employment of fluorescent-based techniques, including microscopy and flow cytometry [23]. We investigated the site of DOX accumulation by tracking the DOX fluorescence and by counter staining the nuclei of MCF-7 cells with Hoechst 33342 (Appendix A). Our results showed free DOX and DOX conjugated SiNPs with different cellular localization, suggesting the exploitation of alternative uptake mechanisms by the cells. DOX conjugated to SiNPs accumulated in vesicle-like organelles, probably by following one or some endocytic pathways, while free DOX was more diffused in cytoplasm and inside the nucleus of cells. A nuclear accumulation was clearly visible, particularly at long times of incubation (Figure 7A). The internalization of doxorubicin, both free and conjugated, was evaluated also by flow cytometry (Figure 7B). We noticed that, while free DOX fluorescence internalized by MCF-7 cells increased during time until reaching almost a plateau at 72 h, for conjugated DOX, after a slight gain at 48 h, there was a reduction in DOX fluorescence, because the cells might implement release/exocytosis mechanisms, as we observed for unconjugated SiNPs in our previous study. At the same time, we also measured NP fluorescence inside the cells by flow cytometry. Histograms in Figure 7C illustrate that both COOH-NPs and DOX-NPs had the same fluorescence intensity after 48 h of incubation at 37 °C, in their respective fluorescence channels, other than that of DOX.

### 2.5. Investigation of Cell Death

The cytotoxic effects of the treatment administered were firstly investigated through Trypan Blue exclusion assay. Appendix A shows cytotoxicity results of MCF-7 cells after incubation with COOH-NPs, DOX-NPs, free DOX or vehicle (DMSO) for 24, 48 and 72 h. We observed that cell viability of MCF-7 after incubation with COOH-NPs was in the range from 85–90% in the longest investigation time compared to control. We hypothesized that the 10–15% decrease in cell viability was to be attributed to the anionic nature of SiNPs unlike those used in our previous study that presented amine functional groups on the shell surface. These differences may arise for different intracellular trafficking of the changed charge-NPs and/or for the different cellular model. The findings reported in the literature concerning the effects of different types of particles functionalized with different chemical groups from numerous research groups did not show a clear trend in terms of cytotoxicity related to the surface charge of SiNPs [24]. Our results differed from what Bhattacharjee and co-workers observed about the role of surface charge in cytotoxicity on NR8383 macrophage cells. Positively charged SiNP-NH2 resulted in more cytotoxicity, reducing mitochondrial metabolic activity and producing high levels of intracellular ROS, while negatively charged SiNP-COOH induced no toxicity [24]. Furthermore, Riebeling et al. evaluated the effects of 40 nm carboxyl-modified polystyrene and 50 nm amine-modified SiNPs on several cell lines and they detected no effect of the former in any test they did, whereas the others displayed cytotoxicity in most cell models [25].

As concerns the effect of DOX-NPs on cell viability, we did not observe a higher cytotoxicity than free DOX, as expected. With free DOX we reached 50% of cell death, in an early investigated time point (24 h), while conjugated DOX showed a slower effect on cell viability (from 75% at 24 h to 30% at 72 h). DOX-NPs are taken into cells via an endocytic process and this requires more time compared to passive diffusion across cell membranes. The nuclear localization of the drug may be important due to its intranuclear site of action and its delivery is required for therapeutic effects. We did not notice DOX fluorescence into the MCF-7 nuclei, but we observed DOX fluorescence in SiNP filled vesicles: this evidence suggests that an alternative site of drug accumulation is represented by lysosomes (as demonstrated by major DOX-NP stock in lysosomes, in comparison to alone SiNPs) (Figure 6C), and this site is able to exert doxorubicin cytotoxic activity. A model of action could be the one described by Wu et al. [26].

As presented by Bar-On et al., DOX is able to arrest MCF-7 cells at G1/S and G2/M checkpoints, whereas it arrests another breast cancer cell line, MDA-MB-231, at G2 /M phase only [27]. The effect of DOX and DOX-NPs on the cell cycle of MCF-7 through PI staining was evaluated on ethanol fixed cells by flow cytometry. Figure 8 shows SiNP-treated cells (with amine functional groups), and a marked increase of MCF-7 cells in G2/M phase after 48h, compared to controls when treated with COOH-NPs (Figure 8A). The same trend of inhibition of cell proliferation was registered also with DOX-NPs. Regarding DOX, our results highlighted a marked decrease of G1 events and concomitant increase in S and G2/M: however, the most relevant feature is the elevated sub-G1 peak, one of the notable hallmarks of apoptosis. MCF-7 incubated with DOX-NPs showed only 4% of cells in sub-G1 phase, approximately six times lower than free DOX-treated cells.

Apoptosis is a well-known process of cell death characterized by specific morphological changes, including cell shrinkage, chromatin condensation and nuclear fragmentation [28]. Micronuclei are chromatin-containing bodies which results from DNA damage and may be considered another feature of apoptosis. Hintzsche et al. presented a higher number of micronuclei in HSC and TK6 cells treated with doxorubicin, related to decrease in cell viability and increase apoptosis per cell [29]. Given the large hypodiploid peak observed in cell cycle analysis and cell death related to DOX treatment, we hypothesized the presence of micronuclei (Figure 7A-iii), resulting from oligonucleosomal DNA cleavage, which also carry the drug molecule intercalated in the DNA. The fluorescence of DOX in Figure 7A-i, in fact, matches with Hoechst staining in the image below, pointed out by small gray triangles. Another parameter to evaluate apoptosis is the exposure on the cell surface of phosphatidylserine from the cytoplasmic side [30]. This is an early characteristic, important to highlight a different apoptotic pattern. Caspase-activated DNase is responsible for the oligonucleosomal DNA degradation during apoptosis. DNA degradation is thought to be important for multicellular organisms to prevent oncogenic transformation or as a mechanism of viral defense. It has been reported that certain cells, including some neuroblastoma cell lines such as IMR-5, enter apoptosis without digesting DNA in such a way [31]. Therefore, we evaluated the exposure of phosphatidylserine (PS), another apoptotic feature, through Annexin V staining by flow cytometry [32]. At an early time point (24 h) we did not detect substantial differences in PS exposure, although DOX-NPs exhibited the highest percentage (Figure 8B,C). Continuing in incubation time, DOX showed a much more pronounced effect, reaching about 75% of Annexin V positive cells. The flow cytometry overlay histogram in Figure 8B, already at 48 h of treatment, well displays the effect of DOX in terms of PS exposure, compared to other treatments and controls.

Lastly, apoptosis is most notably recognised by some morphological features that occur during its progression. Early alterations include cell shrinkage and rounding, because of the retraction of lamellipodia and the breakdown of cytoskeleton [33]. Images obtained under the phase-contrast microscope at higher magnification highlighted some of the main morphological-like apoptotic events, both in MCF-7 incubated with DOX and DOX-NPs, as detailed in Appendix A.

### 2.6. Effect on Tumor Spreading and Invasion

Cell invasion is the main process for cancer progression and depends on several signaling pathways that control cytoskeletal dynamics and the turnover of cell-matrix and cell-cell junctions, followed by cell migration into the adjacent tissues [34]. CD44 is a receptor for extracellular matrix (ECM) components, mainly hyaluronic acid (HA), and a most common cancer stem cell (CSC) marker. Although it is involved in normal physiological events, such as cell-cell interaction and cell-matrix interaction, its high expression has been associated with tumor progression, migration and metastasis. Recently, it has also been linked with anticancer drug resistance, cell proliferation, tumor recurrence and invasion in multiple types of tumors [35,36]. In particular, in breast cancer cell lines several studies reported that CD44+ CSCs have a higher tumorigenicity [37]. Wolf and colleagues, moreover, with the aid of an engineered HA hydrogel-based matrix, have highlighted the importance of CD44 for cell migration and adhesion through the formation of microtentacles on cells [38]. Due to previously detected structures similar to microtentacles and/or long filopodia (inset in Figure 9A) and to the relevance of this surface molecule, we investigated CD44 expression on MCF-7 after treatments. In particular, these membrane formations are tunneling nanotubes (long, thin protrusions used by tumor cells to adhere to and migrate through the extracellular matrix) and migrasomes [39], descending to the ameboid migratory phenotype of MCF-7 [40,41].

The histogram in Figure 9A shows that DOX-NPs induced a gradual decrease of CD44 expression on the cell surface, by flow cytometry, while free DOX-treated cells display a time-dependent opposite trend, according with Fang et al., who observed a dose-dependent increase of CD44 mRNA expression in MCF-7 cells after treatment with DOX at different concentrations [37].

Wang et al. observed that doxorubicin promotes the secretion of CD44+ enriched exosomes from MCF-7 cells, which transfer resistance properties by initiating several intracellular signaling pathways [35,42]. Both in normal physiological, as well as pathological conditions, cells release membrane-bound vesicles such as exosomes, microvesicles and apoptotic bodies. These extracellular vesicles (EVs) could facilitate communication between cells through their cargo such as proteins, miRNA, DNA and mRNA [43,44]. Thus, we analyzed all media collected before the detachment of MCF-7 cells by flow cytometry, adopting Dako CytoCountTM beads (for counting), FSC/SSC Megamix-Plus beads (for appropriate setting) and anti-CD81 monoclonal antibody (for exosome labelling) to investigate the EVs secreted by the cells.

The results show that MCF-7 cells release both EVs (CD81−) and exosomes (CD81+) containing SiNPs (Figure 9B-upper contour plots), as we presented in our previous study in lymphoid and myeloid cells [1]; thus, cells are able to dispose of nanoconstructs, with different intracellular pathways. As depicted in contour plots of Figure 9B at the bottom (gating strategy of negative sample, i.e., w/o CD44 staining in Appendix A) all media contain both CD44+CD81− and CD44+CD81+ vesicles. We observed a higher amount of events in the control sample (data not shown). When these data were related with the corresponding number of cells in the sample (obviously much greater in cells w/o treatment), a higher number of events per cell-unit was detected in DOX-treated samples, as expected; of note, DOX-NPs (Figure 9C) did not show the same profile. This finding highlights, besides an opposite trend in CD44 expression on the cell surface, the lack of the enrichment in CD44+ exosomes, for DOX-NP treated cells. Taking into account the establishment of drug resistance in other cells, neighboring or distant in the body, DOX-NPs may have an advantage in respect of pathology spreading and metastasization processes compared to free DOX treatment.

## 3. Discussion

Our research analyzed, in depth, SiNPs mitochondrial internalization and their response to this stimulus, together with the development of an NP-linked doxorubicin delivery system. Shi et al. presented, for example, the benefits of doxorubicin-conjugated immuno-nanoparticles, a polymeric core-shell NP system where both DOX and antibody targeting ligands were coupled on a NP surface through a simple conjugation chemistry, on SKBR-3 cell line, compared to healthy HMEC-1 cells [45]. Khutale and co-workers developed a multifunctional NP system based on gold NPs (AuNPs), which are extensively used for drug delivery, by exploiting a pH-dependent release of DOX once inside the cells to increase the availability of the drug and avoid its efflux [10]. Aryal et al. constructed pH-responsive Au-NPs conjugated with DOX which have multifaceted potential for future applications to simultaneously enhance computed tomography (CT), imaging contrast or photothermal cancer therapy while delivering anticancer drug to the target [46]. In the present study, fluorescent silica nanoparticles conjugated with DOX were evaluated for their anticancer properties in breast cancer models. The results indicated that DOX-NPs are internalized by cells through endocytic processes and co-localized with lysosomes, revealing their anticancer action, affecting the cell cycle and inducing apoptosis. Indeed, we evaluated the effect on the expression of CD44 antigen, a molecule involved in adhesion, and therefore in cell spreading in tumor invasion. DOX-NPs induced a gradually decrease of CD44 expression on the cell surface, as well as a minor release of CD44+ EVs. As is known, the availability of CD44 at the cell surface provides cancer cells with mechanisms of survival that perpetuate tumor growth. Modulating the levels of CD44 at the cell surface in cancer cells, as well as interaction with hyaluronan, is thus of great importance for cancer therapy, to disrupt the signaling pathways that favor tumor progression and pathology spreading [47]. Our data suggest that the delivery system itself works insofar it allows the internalization of DOX not from passive transport but via endocytic pathways, reaching some main organelles, such as lysosomes and mitochondria. Since drug nanocarriers, once inside the cells, will encounter a series of endocytic compartments with increasing acidity, in future, the employment of some acid-cleavable linkers between DOX and NPs may increase cytotoxicity. The nanosystem will increase the overall intracellular accumulation of the anticancer drug, through the EPR effect; DOX will only be released in a pH-dependent controlled manner, thereby improving the efficacy of the treatment with a great reduction of side effects to normal tissues. Moreover, the addition of a targeting molecule, by exploiting the functional groups on the shell surface, may achieve a specific cellular targeting, to localize and amplify the cytotoxic effect. Alternatively, we observed and analyzed in depth the mitochondrial localization of fluorescent SiNPs in the peri-mitochondrial space and their effects in situ. Although we detected an initial ROS overexpression, mitochondrial respiration was not substantially impaired, thanks to the amount of GSH for ROS buffering. This finding might suggest the use of NPs in combination with other drug treatments to potentiate their cytotoxic effect. Moreover, the SiNPs localization might be of great impact for those human illnesses, including diabetes, neurodegenerative disease such as Parkinson’s and Alzheimer’s, and many types of cancer, linked to mitochondrial dysfunctions. Increasing interest is addressed to mitochondrial targeting for both probes and therapeutics. Thanks to their specific subcellular localization, fluorescent SiNPs might be advantageous in these fields of application.

## 4. Materials and Methods

GSH, dithiobis-(2-nitrobenzoic acid) (DTNB) as well most of the reagent-grade chemicals were purchased from Sigma-Aldrich (Milan, Italy).

### 4.1. Cell Culture

MCF-7 (breast cancer cell line) and U937 (human myelomonocytic cell line) cells (Sigma-Aldrich, St. Louis, MO, USA) were grown at 37 °C in humidified incubator with 5% CO_2_, in Roswell Park Memorial Institute Medium (RPMI) 1640 (Sigma-Aldrich, St. Louis, MO, USA) with stable l-glutamine supplemented with 10% *v*/*v* heat-inactivated fetal bovine serum (FBS; Gibco; Thermo Fisher Scientific, Inc., Waltham, MA, USA) and 1% *v*/*v* penicillin/streptomycin (Sigma-Aldrich, St. Louis, MO, USA).

### 4.2. Isolation of Mitochondria

Mitochondria were isolated by differential centrifugation, as detailed in [48]. In brief, the U937 cells (40–50 × 106) were washed twice in phosphate buffered solution (PBS, 136 mM NaCl, 10 mM Na2HPO4, 1.5 mM KH_2_PO_4_, 3 mM KCl; pH 7.4) and re-suspended (10 × 10^6^ cells/mL) in ice-cold homogenizing buffer (HB, 225 mM mannitol, 75 mM sucrose, 0.1 mM EGTA, protease inhibitor cocktail, 5 mM Tris-HCl, pH 7.4). The cells were homogenized with 30–40 strokes, by using a glass potter placed in an ice-bath. The efficiency of the homogenization process was monitored under the microscope, by counting the number of residual Trypan Blue negative cells. The homogenate was centrifuged at 1000× *g* for 10 min at +4 °C and the supernatant (S1) collected for the final centrifugation. The pellet was re-homogenized and the supernatant (S2) added to S1 and (S1 + S2) centrifuged at 12,000× *g* for 30 min at 4 °C. The pellet, corresponding to the mitochondrial fraction, was carefully resuspended in HB. Aliquots of mitochondrial suspensions were collected to perform further analyses and mitochondrial protein assay. The remaining suspension was centrifuged at 12,000× *g* for 20 min at +4 °C and the mitochondrial pellet processed for the GSH measurement by HPLC, as detailed below.

### 4.3. Measurement of GSH Content in Cells and Mitochondria by High Performance Liquid Chromatography

The determination of GSH content in U937 cells and mitochondria was performed as described in Fiorani et al. [49]. Briefly, the cellular (1 × 10^6^ cells) and mitochondrial pellets were suspended in 100 µL of lysis buffer (0.1% Triton X-100; 0.1 M Na_2_HPO_4_; 5 mM Na-EDTA, pH 7.5), vortexed and kept for 10 min on an ice bath. Thereafter, 15 µL of 0.1 N HCl and 140 µL of precipitating solution (0.2 M glacial meta-phosphoric acid, 5 mM sodium EDTA, 5 M NaCl) were added to the samples. After centrifugation, the supernatants were collected and kept at −20 °C until the HPLC analyses. Just before analysis, 60 µL of the acid extract were supplemented with 15 µL of 0.3 M Na_2_HPO_4_ and 15 µL of a solution containing 20 mg of DTNB in 100 mL of sodium citrate (1% *w*/*v*). The mixture was stirred for 1 min at room temperature and, after 5 min, filtered through 0.22 µm pore micro-filters. The resulting samples were finally analyzed for their GSH content by an HPLC assay [50], using a 15 cm × 4.6 mm, 5 µm Supelco Discovery^®^ C18 column (Supelco, Bellefonte, PA, USA). The UV absorption was detected at 330 nm. The injection volume was 20 µL. The retention time of GSH was approximately 15.7 min.

### 4.4. Flow Cytometric and Confocal Analyses

Isolated mitochondria were further analyzed by Flow Cytometry and Confocal Microscopy. For the determination of mitochondrial potential (ΔΨm) and mitochondrial mass we employed two different probes, respectively, tetramethylrhodamine ethyl ester perchlorate (TMRE) and MitoTrackerTM Green (MTG). MTG (Thermo Fisher Scientific, Waltham, MA, USA) is a mitochondrial-selective fluorescent label which passively spreads through the plasma membrane and accumulates in active mitochondria where it covalently binds to mitochondrial proteins by reacting with free thiol groups of cysteine residues, measuring the mitochondrial mass [51]. MTG 50 nM was added to the samples, incubated for 30 min at 37 °C and then acquired. TMRE (Sigma-Aldrich, St. Louis, MO, USA) 40 nM, which is a ∆Ψm-specific stain able to selectively enter active mitochondria, was added to the samples 15 min before the acquisition [52]. To establish the healthy condition of isolated mitochondria we also used carbonyl cyanide m-chlorophenylhydrazone (CCCP), which is a typical mitochondrial uncoupler, leading to the dissipation of mitochondrial membrane potential [53]. After the incubation of control mitochondria with 10 μM CCCP for 45 min, we observed a decrease of TMRE, resulting in a correct procedure of mitochondria isolation. Thus, we proceeded with TMRE and MTG staining of all samples, with were analyzed with a FACSCantoTM II flow cytometer (BD Biosciences, San Jose, CA, USA), equipped with an argon laser (Blue, Ex 488 nm), a helium-neon laser (Red, Ex 633 nm) and a solid-state diode laser (Violet, Ex 405 nm). For each sample, at least 10,000 events were acquired. Data analyses were performed with FACSDivaTM softwares (BD Biosciences, San Jose, CA, USA). Furthermore, a qualitative analysis of morphological features and the localization of NPs within isolated mitochondria was applied by a Leica TCS SP5 II confocal microscope (Leica Microsystem, Wentzler, Germany) with 488, 543 and 633 nm illuminations and oil-immersed objectives. The images were further processed and analyzed in ImageJ software (National Institutes of Health, Bethesda, MD, USA).

### 4.5. Transmission Electron Microscopy (TEM)

U937 cells were collected and immediately fixed in 2.5% glutaraldehyde for 1 h. They were rinsed with 0.1 M phosphate buffer and postfixed with 1% osmium tetroxide for 1 h. After this, the cells were dehydrated in alcohol of increasing concentration and embedded into epoxy resin at 60 °C. The ultrathin sections (60–80 nm) were collected on 400 mesh nickel grids, stained with uranyl acetate and lead citrate and then detected by TEM. 

### 4.6. DOX-NP Synthesis and Physicochemical Characterization

Fluorescent silica nanoparticles were synthesized as previously described [17] by choosing excitation and emission wavelengths not superimposable to DOX ones (max. absorption 470 nm, max emission 595 nm) [54]. in order to simultaneously track both fluorescent species with a spectrophotometer.

As depicted in Figure 1, the conjugation of doxorubicin (DOX) to SiNPs required different steps. Firstly, we changed the functional group on the NP surface from -NH2 to COOH, in order to bind DOX. 2 mL of NPs were diluted with 222 µL of carbonate/bicarbonate 10X pH 9.5 and 1.2 mL of carbonate/bicarbonate 1X pH 9.5 and 10.9 mg of succinic anhydride (200 eq., 10 mg/mL in DMSO). The mixture was stirred for 1 h at 40 °C, in the dark and then concentrated with Amicon ultra 50 kDa (Merck Millipore, Burlington, MA, USA) and purified by means of size exclusion chromatography SEC, (Sephadex G25, GE Healthcare, Chicago, IL, USA) using water as eluent. The quantification of carboxylic groups on the surface was obtained using a colorimetric indirect method (see detailed protocol in Appendix A). COOH-NPs were dialyzed against 50mM MES pH = 5 for 24 h at 4 °C. 379 µL of NPs-COOH in MES were diluted with 500 µL of 50 mM MES pH = 5. 28.6 µL of 1-ethyl-3-(3-dimetilaminopropil) carbodiimide (EDC, 10mg/mL in DMSO), 21.2 µL N-hydroxysuccinimide (NHS, 10 mg/mL in ddH_2_O), and 0.5 mg of doxorubicin (10 mg/mL in DMSO) (Sigma-Aldrich, St. Louis, MO, USA) were added to the solution. The mixture was stirred for 2.5 h at RT, in the dark. The mixture was concentrated with Amicon ultra 50 kDa (Merck Millipore, Burlington, MA, USA) and purified by means of SEC (Sephadex G25, GE Healthcare, Chicago, IL, USA) using water as eluent. The product was analyzed by Agilent Cary 60 UV-Vis spectrophotometer (Agilent Technologies, Santa Clara, CA, USA), recording the whole spectrum. The efficiency of drug conjugation was then calculated using a direct method by measuring DOX linked to NPs by spectrophotometric analysis as depicted in Appendix A. SiNP physicochemical characterization was achieved as described in our previous study [1] and summarized in Appendix A.

### 4.7. Nanoparticle Uptake and Intracellular Localization

SiNP dispersions and DOX were prepared, diluting the stock solution, previously filtered (0.22 μm membrane filter-Euroclone SpA, Milan, Italy) under a sterile environment, into complete medium at RT (to ensure better NP dispersion), immediately prior to the treatment of cells. To evaluate SiNP uptake and intracellular localization we have to consider each cell line separately.

Details of SiNP uptake experiments on U937 cells, as concentrations and times, are reported in Sola et al., 2021 [1].

For doxorubicin-conjugated NP, we firstly defined DOX IC50 value on MCF-7 cells. Based on the literature, we titrated DOX by exposing cells for 24 h to different concentrations of drug and then assessed cell viability (Trypan Blue exclusion and PI test). We diluted DOX-NPs at the same concentration of drug (1 µg/mL), and, based on the fluorophores concentration of DOX-NPs, we calculated the dilution for COOH-NPs to be used as control. NP dispersions and DOX were prepared by diluting the stock solution, previously filtered with a 0.22 μm membrane filter (Euroclone SpA, Milan, Italy) under a sterile environment, into complete medium used for cell culture at room temperature (to ensure better NP dispersion), immediately prior to the treatment of cells. MCF-7 (105 cells/well) were seeded in MatTek glass bottom chambers (MatTek Ashland, MA, USA) and incubated for 24 h at 37 °C in a humidified incubator with 5% CO_2_. Then, the medium was removed and replaced with fresh medium containing free DOX (1 µg/mL), DOX-NPs (1 µg/mL of DOX) and COOH-NPs (diluted 1:50) and incubated for 4 h. The conditioned media were removed and cells were then washed twice with PBS 1X (10 mM NaPi, 150 mM NaCl, pH 7.3–7.4) and stained with different cellular markers. 100 nM of the acidotropic dye LysoTrackerTM Green (LTG) (Thermo Fisher Scientific, Waltham, MA, USA) incubated for 45 min was used to investigate the involvement of the lysosomal compartment [55]. Mitochondrial features were investigated by tetramethylrhodamine ethyl ester perchlorate (TMRE) (Sigma-Aldrich, St. Louis, MO, USA) 40 nM, a ∆Ψm-specific stain able to selectively enter active mitochondria, was added to the samples 15 min before the acquisition [52]. Confocal microscopy analyses were applied by a Leica TCS SP5 II confocal microscope (Leica Microsystem, Wentzler, Germany) with 488, 543 and 633 nm illuminations and oil-immersed objectives. The images were further processed and analyzed in ImageJ software (National Institutes of Health, Bethesda, MD, USA). Colocalization analyses were performed using JACoP plugin in ImageJ software (NIH, Bethesda, MD, USA) [21]. Pearson’s correlation coefficient (PCC) was used as the parameter to measure colocalization in our samples.

For quantification of NPs internalized into cells through flow cytometry, 105 cells/well were seeded in 12-well plates and incubated for 24 h at 37 °C with 5% CO_2_, to let them adhere. The MCF-7 cells were then treated and incubated for different time points (24, 48 and 72 h). After trypsinization cells were harvested, centrifuged at 1200 rpm for 5 min, washed twice with PBS 1X and analyzed with NovoCyte^®^ 3000 flow cytometer (ACEA Biosciences, San Diego, CA, USA), equipped with three lasers (Violet Ex 405 nm, Blue Ex 488 nm and Red Ex 640 nm) and with a FACSCantoTM II flow cytometer (BD Biosciences, Franklin Lakes, NJ, USA), equipped with an argon laser (Blue, Ex 488 nm), a helium-neon laser (Red, Ex 633 nm) and a solid-state diode laser (Violet, Ex 405 nm).

At least 10,000 events were acquired for each sample. Data analyses were performed with NovoExpress Software, with Kaluza Analysis 2.1 (Beckman Coulter, Brea, CA, USA) and FACSDivaTM softwares (BD Biosciences, San Jose, CA, USA).

### 4.8. DOX Internalization and Accumulation

MCF-7 (4.5 × 10^4^ cells/well) were seeded on Millicell EZ slides (Millipore, Burlington, MA, USA) and grown for 24 h at 37 °C in 5% CO_2_ incubator. The medium was removed and replaced with fresh medium containing free DOX (1 µg/mL), DOX-NPs (1 µg/mL of DOX), COOH-NPs (diluted 1:50) and DMSO (vehicle control). At the different time points of 24, 48 and 72 h, the treatment was gently aspirated from the left corner of each well and the cells washed twice with PBS 1X. 200 µL of 14 µM Hoechst 33342 (Thermo Fisher Scientific, Waltham, MA, USA) was added to each well and incubated for 10 min in the dark. Finally, after the incubation, the slides were washed with PBS 1X and mounted with nail polish on microscope slides. Cells were immediately visualized with a Nikon Optiphot 2 fluorescence microscope (Nikon, Tokyo, Japan) equipped with a CCD camera, using DM580 and Omega optical INC. XF 1009/25 filters at 20×, 40× and 100× oil immersion magnification. The images were collected using NIS-Elements (Nikon, Tokyo, Japan) software.

For fluorescence quantification of DOX inside cells through flow cytometry, 105 cells/well were seeded in 12-well plates and incubated for 24 h at 37 °C with 5% CO_2_, to let them adhere. The MCF-7 cells were then treated and incubated for different time points (24, 48 and 72 h). After trypsinization, cells were harvested, centrifuged at 1200 rpm for 5 min, washed twice with PBS 1X and analyzed with a FACSCantoTM II flow cytometer (BD Biosciences, San Jose, CA, USA), equipped with an argon laser (Blue, Ex 488 nm), a helium-neon laser (Red, Ex 633 nm) and a solid-state diode laser (Violet, Ex 405 nm). For each sample, at least 10,000 events were acquired. Data analyses were performed with Kaluza Analysis 2.1 (Beckman Coulter, Brea, Ca, USA) and FACSDivaTM software (BD Biosciences, San Jose, CA, USA).

### 4.9. Cytotoxic Activity

#### Trypan Blue Exclusion Assay

The cytotoxicity of DOX and DOX-NPs on MCF-7 cells was determined by Trypan Blue exclusion assay, as described by Strober [56]. 105 cells/well were seeded in 12-well plates and incubated for 24 h at 37 °C with 5% CO_2_, to let them adhere. The MCF-7 cells were then treated with free DOX (1 µg/mL), DOX-NPs (1 µg/mL of DOX) and COOH-NPs (diluted 1:50) and incubated for different time points (24, 48 and 72 h). Controls were incubated with fresh medium and, moreover, with DMSO as vehicle control. After trypsinization, an aliquot of cell suspension was mixed with an equal volume of 0.4% Trypan Blue solution (Life Technologies Corporation, Eugene, OG, USA), incubated for approx. 3 min at room temperature and then counted with a Bürker chamber. Cells were visually examined, with light microscopy (Nikon Corporation, Tokyo, Japan), to determine whether cells took up or excluded dye: viable cells typically show a clear cytoplasm (unstained) whereas nonviable cells show a blue cytoplasm (stained). The percentage of viable cells was calculated, related to control cells, as usual.

### 4.10. Evaluation of Cell Surface Expression of Phosphatidylserine

To investigate cell surface expression of phosphatidylserine (PS), a feature of programmed cell death, an aliquot of cells previously treated, was stained with FITC-conjugated Annexin V (Immunostep, Salamanca, Spain). Washed cells were centrifuged at 1200 rpm for 5 min and the pellet resuspended in 50 µL of Annexin V Binding Buffer (10 mM Hepes/NaOH, pH 7.4; 140 mM NaCl; 2.5 mM CaCl_2_). Finally, 2.5 µL of Annexin V-FITC were added to the cell suspension, incubated for 15 min at room temperature and analyzed by flow cytometry.

### 4.11. Morphological Changes

For the analysis of morphological changes, MCF-7 cells were seeded in 12-well plates and incubated for 24 h, then treated with free DOX (1 µg/mL), DOX-NPs (1 µg/mL of DOX), COOH-NPs (diluted 1:50) and DMSO (vehicle control) and incubated for different time points (24, 48 and 72 h). Finally, the cells were observed under bright field inverted light microscope Nikon eclipse TS100 (Nikon, Tokyo, Japan) at 10× and 40× magnification, to highlight, respectively, the detachment of cells and cell density and morphological changes related to cell death.

### 4.12. Cell Cycle Analysis

The distribution of DNA in the cell cycle was investigated by flow cytometry. MCF-7 (5 × 10^5^ cells/well) were seeded in 6-well plates, grown for 24 h and, finally, exposed to a solution of free DOX (1 µg/mL), DOX-NPs (1 µg/mL of DOX), COOH-NPs (diluted 1:50) and DMSO (vehicle control) and incubated for different time points (24 and 48 h). After trypsinization cells were harvested, centrifuged at 1200 rpm for 5 min and washed twice with PBS 1X. The pellets were fixed drop by drop with ice-cold ethanol (70%) and stored at +4 °C, until the analyses. For cell cycle analyses, samples were washed twice with PBS 1X and each pellet was resuspended in 440 µL of PBS 1X, to which 10 µL of 1 mg/mL PI (Sigma-Aldrich, St. Louis, MO, USA) and 50 µL of 1 mg/mL RNAse (Sigma-Aldrich, St. Louis, MO, USA) were added, to reach a final volume of 500 µL. The samples were well resuspended and incubated at 37 °C for at least 30 min or at +4 °C until analysis by flow cytometry.

### 4.13. Flow Cytometric Analysis of CD44 Expression

To evaluate CD44 expression on cells, fluorochrome-conjugated monoclonal antibodies were added to 50 µL of cell pellets. Mouse anti-human antibody anti-CD44 RPE (clone J.173) (Beckman Coulter, Brea, CA, USA) was added at dilutions according to the manufacturer’s instructions. After 15 min of incubation at RT in the dark, samples were acquired by flow cytometry.

Media of MCF-7 cells, which have undergone the different treatments, were analyzed to study the presence of extracellular vesicles released by cells [57]. 300 μL of medium for each sample was carefully dispensed at the bottom of the tube and 25 μL of Dako CytoCountTM beads (Thermo Fisher Scientific, Waltham, MA, USA) was added. Megamix-Plus beads (Biocytex, Marseille, France) were acquired according to manufacturer’s instructions to set up FSC and SSC parameters for EV detection [58]. Supernatants were also stained with mouse anti-human antibody anti-CD81-FITC (clone JS-81) (BD Biosciences, San Jose, CA, USA), a tetraspanin frequently identified in exosomes and considered classical markers of exosomes [59]. Moreover, media were stained with anti-CD44 RPE (clone J.173) (Beckman Coulter, Brea, CA, USA), to evaluate the expression of this tumoral marker, not only on the cell surface, but also on the EVs released from cells. After 15 min of incubation at RT in the dark, samples were acquired by flow cytometry.

### 4.14. Statistical Analysis

Data are shown as mean (or percentage, as indicated) ± standard deviation (sd) of at least three independent experiments, when carried out; otherwise, two replicates of experiments were run. The means of two groups were compared by using a paired T test or by one-way ANOVA, followed by a Bonferroni post-hoc test. The p values less than 0.05 were considered statistically significant. All statistical analyses were performed using GraphPad Prism 5 (GraphPad software, San Diego, CA, USA).

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
