# Peer review of "Fluorescent Silica Nanoparticles Targeting Mitochondria: Trafficking in Myeloid Cells and Application as Doxorubicin Delivery System in Breast Cancer Cells"

_ijms, 2022, doi:10.3390/ijms23063069_

Round 1
Reviewer 1 Report
The manuscript provides information regarding the cytotoxic effects of DOX-NPs on breast cancer cell line MCF-7. The authors used cutting edge methodology for elucidating the mechanism of action of DOX-NPs which is of great importance for cancer therapy. The paper is generally well written and structured.
Author Response
Reviewer 1
The manuscript provides information regarding the cytotoxic effects of DOX-NPs on breast cancer cell line MCF-7. The authors used cutting edge methodology for elucidating the mechanism of action of DOX-NPs which is of great importance for cancer therapy. The paper is generally well written and structured.
The Authors would like to thank the Reviewer regarding his/her comments.
Reviewer 2 Report
This is a very interesting work on fluorescent silica nanoparticles targeting mitochondria and their potential use for drug (doxorubicin) delivery to cancer cells. The methods used are very diverse and up to date and they are used in the best well validated way. The results seem exciting and they are clearly described and explained and the figures are informative enough.
There are minor mistakes in English in several places throughout the text (as an example the use of analysis and analyses with the tense of the verbs or the use of proverbs instead of adjectives Gradually in the place of gradual) One sentence in particular on lines 445-448 doesn't make any sense.
What needs rewriting are the conclusions where mainly the work of other studies are presented and not the actual conclusions of this paper or a discussion in comparison to the other studies. Such an extensive work with so many interesting results needs clear conclusions.
Author Response
Reviewer 2
This is a very interesting work on fluorescent silica nanoparticles targeting mitochondria and their potential use for drug (doxorubicin) delivery to cancer cells. The methods used are very diverse and up to date and they are used in the best well validated way. The results seem exciting and they are clearly described and explained and the figures are informative enough.
The Authors would like to thank the Reviewer regarding his/her comments and suggestions to improve the manuscript.
There are minor mistakes in English in several places throughout the text (as an example the use of analysis and analyses with the tense of the verbs or the use of proverbs instead of adjectives Gradually in the place of gradual) One sentence in particular on lines 445-448 doesn't make any sense.
The sentence on lines 445-448 was changed, as well as the caption of Fig. 9. English minor mistakes are now adjusted.
What needs rewriting are the conclusions where mainly the work of other studies are presented and not the actual conclusions of this paper or a discussion in comparison to the other studies. Such an extensive work with so many interesting results needs clear conclusions.
We rewrote the final discussion considering our experimental data, on the basis the reviewer’s suggestions